

# The Doppler wind, temperature, and aerosol RMR lidar system at Kühlungsborn/Germany – Part 1: technical specifications and capabilities

Michael Gerding[1], Robin Wing[1], Eframir Franco-Diaz[1], Gerd Baumgarten[1], Jens Fiedler[1], Torsten Köpnick[1], and Reik Ostermann[1]

[1]Leibniz Institute of Atmospheric Physics at the University of Rostock, Kühlungsborn, Germany

**Correspondence:** Michael Gerding (gerding@iap-kborn.de)

**Abstract.** This paper describes the technical specifications of the extensions made to the middle atmospheric lidar facility at the Leibniz Institute of Atmospheric Physics in Kühlungsborn, Germany (54.12°N, 11.77°E). The upgrade complements the existing, vertically pointing daylight-capable Rayleigh-Mie-Raman (RMR) temperature lidar with a 2-beam, nighttime-only RMR wind-temperature lidar. The 2-beam system comprises an independent lidar with laser, telescopes, and detectors, which is synchronized with and adapted to the temperature lidar. This work intends to highlight the recent innovations in the construction of a 3-beam Doppler-Rayleigh wind lidar system using the single-edge Iodine-cell technique, which allows for the simultaneous measurement of wind, temperature, and aerosols. We will detail supporting subsystems that allow for a high degree of lidar automation and concisely provide key technical information about the system that will support readers in the development of additional Doppler-Rayleigh wind lidar systems. We show an example of time-resolved temperature and wind soundings reaching up to ∼90 km. These data agree well with ECMWF-IFS profiles between 35 and ∼50 km but show a much larger variability above. In the companion article, we will present the algorithm design and uncertainty budgets associated with the data processing chain.

## 1 Introduction

Simultaneous, common volume measurements of both temperature and wind are essential for understanding the wave-driven dynamics and circulation of the middle atmosphere (Andrews et al., 1987). This kind of coupled temperature-wind measurement is extremely rare in the middle atmosphere, despite its importance to fundamental studies of wave dynamics, energy transfer, and turbulence (Lübken et al., 1993; Wing et al., 2021), as well as studies of synoptic scale phenomena like the Quasi-Biennial Oscillation (QBO) (Baldwin et al., 2001) and Polar Vortex dynamics (Manney et al., 1999).

This fundamental need for observations has become more imperative as modern versions of atmospheric reanalysis push higher into the middle atmosphere and try and resolve smaller grid scales (Dee et al., 2011). The pressing need for observations to help validate and constrain atmospheric reanalysis and improve weather forecasting has provided the impetus for large European-level projects, such as ARISE, aimed at measuring temperature and wind in the middle atmosphere (Blanc et al., 2019). Ground-based lidar serves an essential role as an independent, unassimilated dataset with which the physics and veracity





of an atmospheric reanalysis can be tested (Marlton et al., 2021). In addition to validation studies, the very high spatial and
temporal resolution of lidar measurements allows us to conduct "discovery science" by measuring unique phenomena such as
multi-stage vertical coupling by gravity waves predicted by theory and modelling (Vadas et al., 2023).

## 1.1 Techniques to Measure Wind in the Middle Atmosphere

At present, there are a limited number of measurement techniques available for the assessment of wind in the middle and
upper atmosphere. Balloon-borne instruments can make in-situ wind and temperature measurements in the troposphere and
lower stratosphere (surface to balloon bursting height at approximately 35 km) (Houchi et al., 2010). Radars are capable
of measuring winds below 20 km and again in the mesosphere between 75 and 100 km due to the presence of favourable
scattering targets (Hocking, 1997) but, are unable to measure in the intervening region. Wind measurements relying on the
release of tracers are also possible in the upper middle atmosphere using rockets (Larsen, 2002; Müllemann and Lübken, 2005).
Unfortunately, these high-quality measurements are expensive and infrequent. Microwave radiometers are capable of making
low-resolution measurements of zonal wind on five pressure levels between 30 and 80 km (**?**). In the upper mesosphere and
lower thermosphere, (UMLT) metal resonance lidars can measure temperature and wind in the so-called metal layer (She and
Yu, 1994; Höffner and Lautenbach, 2009). Finally, there were several past satellite instruments which measured wind at various
levels. In the stratosphere and UMLT missions like WINDII (Gault et al., 1996), TIDI (Killeen et al., 1999), HRDI (Swinbank
and Ortland, 2003), and more recently ICON (Immel et al., 2018) used passive remote sensing to measure horizontal winds.
In the troposphere and lower stratosphere, the AEOLUS mission used active remote sensing and the double-edged Doppler-
Rayleigh technique to measure line-of-sight winds (Stoffelen et al., 2005).

## 1.2 Other Doppler-Rayleigh Wind lidars

The pressing need to measure middle atmospheric winds, which has motivated the construction of previous Dopper-Rayleigh
wind lidars, has been discussed in the literature (Baker et al., 2014). There are currently four other operational stations in
the world: (1) L'Observatoire de Haute Provence (OHP) in France (44°N, 6°E), (2) the Arctic Lidar Observatory for Middle
Atmosphere Research (ALOMAR) in Norway (69°N, 16°E), (3) Observatoire de Physique de l'Atmosphere de la Réunion
(OPAR) on La Reunion island (21°S, 55°E), and (4) two mobile wind lidars in China developed in Hefei region (approx.
32°N, 117°E). In addition to these permanent stations, short-lived Doppler-Rayleigh lidars have been built at the University
of Michigan (Fischer et al., 1995), Arecibo Observatory (Tepley, 1994; Friedman et al., 1997), and NASA Goddard (Gentry
et al., 2000). A Doppler wind lidar using aerosol scattering was recently described by (Mense et al., 2023). To provide a
comprehensive overview of the current 'state of the art', we will give a detailed review of the technical developments of each
station and summarize the key scientific results that have been obtained using Doppler-Rayleigh lidars.





### 1.2.1 Observatoire de Haute Provence (OHP)

The first Doppler-Rayleigh wind lidar was constructed in 1989 at OHP and was capable of measuring winds from 25 to 60 km
with a vertical resolution of 2 km and a temporal resolution of 2 hours (Chanin et al., 1989). This lidar used a doubled Fabry-Pérot interferometer (FPI) in a temperature-controlled cavity. The system was calibrated by routinely taking zenith-pointing measurements throughout the night. During these vertical measurements, the two slits of the interferometer were recentred about the laser emission wavelength by changing the interferometer cavity temperature. The underlying assumption is that the integrated vertical wind over the atmospheric column is very small and the resulting Doppler shift is negligible. Following
the interferometer calibration in the zenith position, off-zenith measurements of Doppler shift can be made by comparing the signal passing through each side of the interferometer. This double-edge Doppler-Rayleigh lidar has the advantage of being technically straightforward, robust over decades, and does not require very fine control over the laser emission wavelength. The disadvantage of the technique is that the instrument function of the double FPI biases the lidar photon counts profile so that it is no longer directly proportional to the density at all altitudes. This is not a problem for wind retrieval, which exploits the ratio
of light passing through both slits and allows altitude-dependent biases to cancel out, but, the lidar profile cannot subsequently be used to derive simultaneous temperatures or aerosols.

Following the eruption of Mount Pinatubo in 1991, it became clear that the Mie scatter influence due to aerosols needed to be addressed in the Doppler-Rayleigh technique (Garnier and Chanin, 1992). In 1999, the OHP wind lidar was upgraded with a higher finesse FPI with pressure regulation, external frequency seeding of the power laser, and a new telescope assembly
(Souprayen et al., 1999b, a). The objective of these upgrades was to extend the wind profiles below 25 km down to 5 km by excluding the aerosol Mie peak. Extensive modelling and calibration work was also done for the FPI and wind retrieval.

An attempt was made to use the wind lidar to explain thin ozone laminae in terms of fully resolved high-frequency gravity waves (Gibson-Wilde et al., 1997). A hodograph technique was used to estimate gravity wave characteristics from the wind lidar and were compared to colocated measurements from the OHP ozone lidar (Godin-Beekmann et al., 2003). The results
of the study were inconclusive. Further gravity wave work using the wind lidar was conducted by Hertzog et al. (2001). This paper contains the first climatology of gravity wave kinetic energy density and spectra in the lower stratosphere made by Doppler-Rayleigh lidar.

Most recently, the OHP wind lidar participated in the Cal/Val program for the AEOLUS satellite. Intercomparisons over a few months were conducted between the OHP wind lidar, AEOLUS, locally launched radiosondes, and ECMWF-IFS winds.
The OHP lidar was in good agreement with the radiosondes (+0.1 ± 2.3 m/s) and with AEOLUS (+1.5 ± 3.2 m/s) (Khaykin et al., 2020). This was among the first confirmations of the space-born wind lidar.

### 1.2.2 Arctic Lidar Observatory for Middle Atmosphere Research (ALOMAR)

The original Doppler-Rayleigh lidar at ALOMAR was constructed in 1994 for temperature, aerosol, and wind soundings in the middle atmosphere. From the beginning, it made use of two lasers emitting at 1064, 532, and 355 nm, and two telescopes
(Von Zahn et al., 2000). Daylight suppression was implemented using double FPIs for UV and visible wavelengths and a



single FPI for IR measurements. The laser was frequency stabilized on a molecular absorption line of iodine. The Doppler measurement system was unique, using a 24-channel ring anode detector which directly imaged the fringes from the double FPI (Rees et al., 1996). This allowed for wind measurements up to ∼25 km during day and night (Baumgarten et al., 1999). Middle atmosphere soundings have been possible day and night for temperatures and aerosols (Noctilucent clouds, NLC)
(Von Zahn et al., 2000). In the following years, continuous improvements and extensions have been made to the lasers and especially the detection bench.

A significant technological development in Doppler-Rayleigh lidars was the development of a pure "single edge" technique, called Doppler Rayleigh Iodine Spectrometer (DORIS), which replaces the fringe measurements in favour of a second iodine gas cell in the detector (Baumgarten, 2010). By basing the entire technique on a single molecular absorption feature in the iodine
spectrum, the frequency and spectral shape of the laser line and lidar signal can be known at every stage of the experiment. This new single-edge technique no longer requires the vertical calibrations of the double-edge FPI technique, can operate in daylight conditions, and simultaneously retrieve temperature, wind, and aerosol parameters without the instrument transmission function effects of the FPI. The increase in performance and capability allowed the lidar to measure winds into the upper mesosphere. These advancements into the mesosphere have allowed for the combination of temperature and wind measurements from
Doppler-Rayleigh and Na resonance lidars to create a unified temperature and wind field from 30 to 110 km, allowing for the first continuous observations of gravity waves from the lower stratosphere to the upper mesosphere and lower thermosphere (Hildebrand et al., 2012). Until now, the ALOMAR RMR lidar using the DORIS technique is the world's only lidar for wind measurements in the mesosphere during daytime (Baumgarten et al., 2015).

Following the implementation of the new DORIS technique, two inter-comparison studies were conducted to systematically
compare the ALOMAR lidar temperatures and winds to rocket, radiosonde, meteor radar, and microwave radiometer measurements, as well as model and reanalysis outputs from ECWMF-ERA5, ECMWF-IFS, MERRA2, and SD-WACCM (Lübken et al., 2016; Rüfenacht et al., 2018). As stated earlier in this section, these inter-comparison exercises are crucial under the European framework for the measurement and forecasting of the middle atmosphere (Blanc et al., 2019). The rocket intercomparisons showed no significant bias between in situ and lidar winds, with an RMS uncertainty of $5 − 7$ m/s for the zonal
winds and $3 − 9$ m/s for the meridional winds. There was good agreement between the radar, lidar, and microwave radiometer, however, substantial biases between observations and reanalysis/models were seen above 0.3 hPa (∼57 km).

Proceeding from a technically sound and well-validated set of lidar observations of temperature and wind, Strelnikova et al. (2020) was able to derive an algorithm to extract the vertical wavelength, intrinsic period, horizontal wavelength, and horizontal phase speed from gravity wave packets in lidar observations. Fully separating the upwards and downwards wave propagation,
vertical profiles of kinetic and potential energy as well as profiles of momentum flux were systematically calculated for the first time in the stratosphere. This advancement in gravity wave measurement and characterization is only possible due to the preceding decades of Doppler-Rayleigh lidar system development and calibration.



### 1.2.3 Observatoire de Physique de l'Atmosphère de la Réunion (OPAR)

The Doppler-Rayleigh wind lidar, located at the high-altitude Maïdo observatory on the island of la Réunion, was constructed
in 2012. This wind lidar operates at an altitude of 2200 m above sea level and is the only Doppler-Rayleigh wind lidar in the
Southern Hemisphere. The OPAR wind lidar is closely based on the OHP wind lidar. The laser, interferometer and optics are
identical to those used at OHP. The receiver assembly is unique and consists of a single 60 cm telescope which rotates through
three fixed positions to measure the zonal and meridional lines of sight as well as the vertical for reference. Despite the smaller
telescope, the OPAR wind lidar has signal levels comparable to the OHP lidar due to its advantageous position on the volcano.
Recently, the station was involved in a long-term validation experiment with the AEOLUS satellite. To date, the observed bias
in the AEOLUS winds is recorded as -1.12 ± 6.49 m/s (50 % larger variance than expected) with significant altitude-dependent
artefacts present in the AEOLUS wind profile (Ratynski et al., 2022). These findings are crucial for the development of the
next generation of space-born wind lidars.

### 1.2.4 Wind lidars in China

In 2012, the first Doppler-Rayleigh lidar was constructed in Asia by the University of Science and Technology of China. This
lidar was mobile and used an Iodine gas cell frequency stabilized tripled Nd:YAG as a transmitter and a double-edged FPI as a
detector. The system was capable of measuring winds from 8 to 40 km and was in good agreement with ECMWF winds below
25 km (Xia et al., 2012). In 2014, the system was upgraded to allow for simultaneous observations of wind and temperature
from 15 to 60 km, however, contamination due to aerosols appears to remain a problem below 27 km (Dou et al., 2014; Zheng
135 et al., 2018).

The first gravity wave work involving this lidar was conducted in 2016. The authors exploited the simultaneous measure-
ments of bandpass filtered temperature and wind perturbations to characterize matching gravity wave phase patterns (Zhao
et al., 2016). Using a hodograph technique, Zhao et al. (2017) were able to distinguish between quasi-stationary mountain
waves and inertial gravity waves at three sites in China.

140 In 2017, the mobile Doppler-Rayleigh lidar was upgraded from a double-edge detection system using an FPI to a single-
edge detection system based on an iodine cell similar to the ALOMAR lidar in 2010. This upgrade allowed for the retrieval of
simultaneous winds and temperatures from 30 to 70 km (Yan et al., 2017).

In 2023, a second Doppler lidar was built at the Anhui Institute of Optics and Fine Mechanics in Hefei, China. This lidar is
unique in that it is built on a rotating turn-table, which permits active rotation during the lidar acquisition. The lidar is based
145 on the double-edge FPI technique and demonstrates excellent wind measurements from 10 km to 30 km (Chen et al., 2023).

## 2 System Design of the New Doppler-RMR at Kühlungsborn

The three-beam RMR lidar at Kühlungsborn comprises two connected lidar systems. The first part of the system is a vertically
emitting, daylight-capable temperature lidar (called 'RMR2' here). This lidar is described in detail by Gerding et al. (2016)



and has been updated recently to integrate with the new system. The second part of the new lidar is a two-beam tiltable system
intended for wind and temperature measurements (called 'RMR3' here). The wind lidar concept follows the DORIS technique
of the ALOMAR RMR lidar (Baumgarten, 2010), i.e. the single-edge technique of using an iodine cell as a spectroscopic
element. We will concentrate here on the RMR3 lidar, but give an update on the vertical pointing lidar and explain the interplay
of both systems.

Our objective for the new system was to design and build a modern Doppler-RMR lidar with a high degree of automation,
which can operate semi-autonomously for the next decades. The system must be able to run without operator intervention,
self-correct common minor problems, and safely shut down in the case of low signal (i.e. clouds) or message an operator
in the case of more serious problems. The new lidar should fit into the existing lab infrastructure at our main building at
Kühlungsborn, separating the laser lab, telescope room, and detection room. An operator, if needed at all, shall be able to
handle the system from anywhere with only a notebook PC and an internet connection. The use of distributed labs favours
sharing the control to different PCs. We decided to combine different system architectures and operating systems, as described
below. This allows for having specialised, independent subunits for, e.g., laser control, beam stabilisation, data acquisition
etc. All subunits communicate via a common network protocol (MQTT: Message Queuing Telemetry Transport) and are
coordinated by a high-level control software called KLAUS (Kühlungsborn Lidar AUtomation System). In the following, we
give a technical overview and describe the relevant subunits and the coordinating software.

A seeded Nd:YAG laser is emitting the first harmonic with a 100 Hz repetition rate. The seed laser is shared with the Nd:YAG
laser of the daylight temperature lidar and stabilized to an iodine absorption line. An etalon-based laser pulse spectrometer
monitors the frequency offset of the pulsed laser. The laser emission is separated pulse-by-pulse with a galvanometer scanner
mirror and directed via two beam-guiding chains to two different telescopes. Cameras attached to the telescopes are used
together with motorised mirrors for automated beam stabilisation. The received light is guided by optical fibres to a single
detection system. Here, again, a galvanometer scanner mirror is used for alternating feed of the light from the two fibres into
the same detector. A photon counting system records the signal of gated avalanche photodiodes on a single-pulse basis.

Figure 1 gives a room-by-room overview of the hardware network: 1) the 'Laser Room RMR2' contains the seed laser
system for both lidars RMR2 and RMR3, and the power laser of RMR2 (not shown), 2) the 'Laser Room RMR3' contains
the power laser for off-zenith beams, laser control system, and laser pulse spectrometer, 3) the 'Telescope Room' contains the
motorized telescopes, motorized beam guiding mirrors, and beam stabilization cameras, and 4) the 'Detector Room' contains
the photodetectors, data acquisition system, and central trigger system. Describing complex networks can be challenging when
viewed in the physical schematic. In the following subsections, we will present a more detailed view of the system by focusing
on one subsystem at a time.



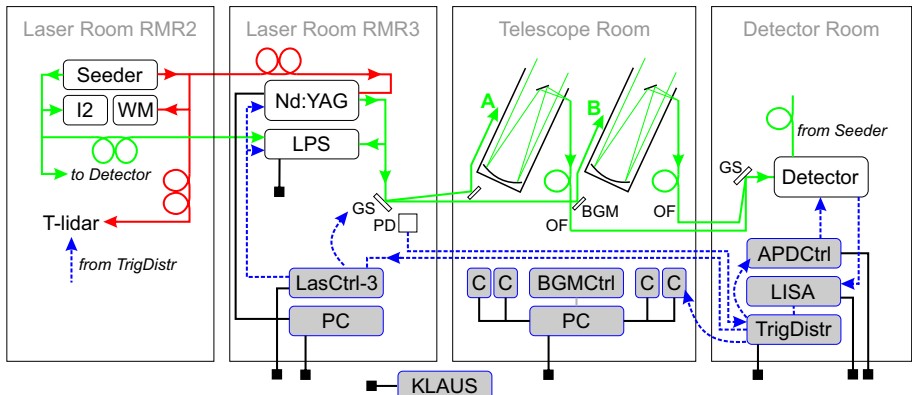

**Figure 1.** Schematic of the new wind lidar system 'RMR3' at Kühlungsborn. I2: Iodine cell, WM: wavemeter, LPS Laser Pulse Spectrometer, GS: galvanometer scanner, PD: photodiode, LasCtrl-3: Laser Controller RMR3, BGM: Beam Guiding Mirror, OF: optical fibre, C: camera, BGMCtrl: BGM Controller, APDCtrl: Avalanche Photo Diode Controller, LISA: LIdar Singleshot Aquisition, TrigDistr: Trigger Distributor, KLAUS: Kühlungsborn Lidar Automation System. A, B denote outgoing beams and telescopes. Green/red: light path of 532/1064 nm light; black: Ethernet/TCPIP connections, grey: USB connection; blue dashed: (coax) data cable. Only the most relevant parts and connections are displayed.

## 2.1 Laser Bench Optoelectronics

We describe in this section the transmitter of our RMR wind lidar, namely the pulsed power laser and the beam guiding on the laser bench, the seeding system as well as the Laser Pulse Spectrometer (LPS) for frequency monitoring of the pulsed laser. The central part of the transmitter is an Innolas SpitLight EVO IV diode-pumped Nd:YAG with an externally stabilized seed laser (see below). The power laser is triggered at 100 Hz by the central trigger controller (see Appendix D). We have developed a cavity control that adjusts the Piezo-coupled end mirror of the laser for build-up time reduction (BUTR). The laser contains 185 a second harmonic generator to get light at 532 nm wavelength with a pulse energy of ∼500 mJ. Only this second harmonic is used, while the remaining IR light is dumped.

Doppler-Rayleigh winds and daylight temperatures require a high degree of stabilization of the power laser frequency. We achieve this by externally seeding the power laser with a Coherent Prometheus 100 laser. The seed laser is tuned to the line 1109 in the iodine absorption spectrum using an $I_2$ vapour cell, i.e. to 532.112 nm (air) (532.260 nm (vac.)). This cell is heated 190 to 45°C. We split some of the frequency-doubled seed light and send 70% through the iodine cell and into a photodiode, and 30% directly to a photodiode. By calculating the ratio of the signal from the two photodiodes, we measure the absorption of the iodine gas at a given wavelength. Stabilization is achieved using built-in cavity control of the seed laser. The $I_2$ line 1109 has been selected for optimal wind sensitivity within the wavelength range that is covered by the Nd:YAG laser. See Fig. 7 below for the measured spectrum of the seeder iodine cell.

The path of the pulsed laser is displayed in Fig. 2. Two cascaded wedge plates (BP) pick ∼0.2% of the pulsed laser for frequency measurements in the LPS (see below). Next, the galvanometer scanner separates the laser pulse-by-pulse into two



different paths. By this, we achieve two virtual 50 Hz subsystems (called A and B) for wind measurements in two directions. The light along path A then encounters Mirror A and is sent to Beam Widening Telescope A (BWT-A), the same with path B. The BWT is a custom-design, 10x telescope. The outgoing pulse is measured by photodiodes (PDs), generating a trigger pulse which is then sent to the central trigger controller. Finally, the light encounters the first of three Beam Guiding Mirrors (BGM-1A/B). This mirror is used to steer the beam into the telescope hall. Technical details of the optics can be found in Appendix A.

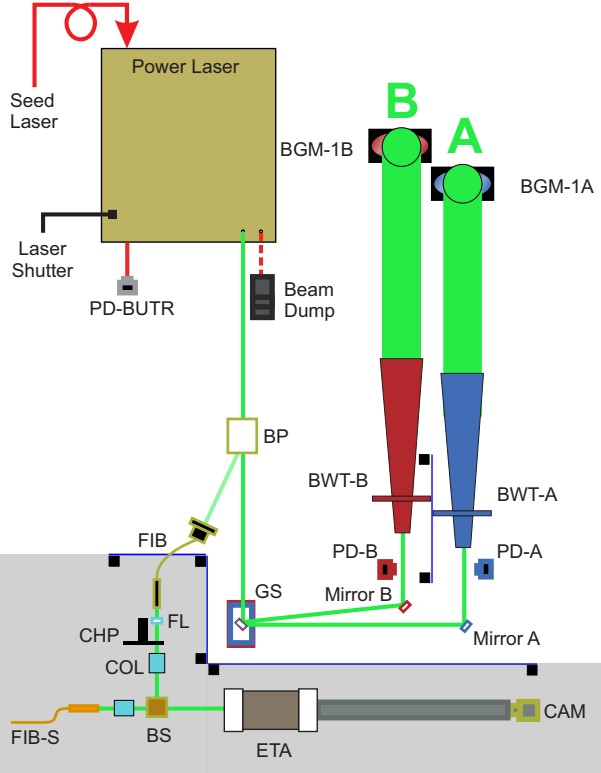

**Figure 2.** Schematic of the transmitting bench. BP: beam picker box, FIB: fibre, FL: focussing lens, CHP: chopper, COL: collimator, FIB-S: seeder fibre, BS: beam splitter, ETA: etalon, CAM: camera, GS: galvanometer, PD: photodiode, BWT: beam widening telescope, BGM: beam guiding mirror. Grey shaded is the Laser Pulse Spectrometer (LPS).

The frequency offset between the power laser and the seed laser as well as the associated jitter between the two frequencies are major sources of systematic uncertainties in wind measurement. In the retrieval of horizontal wind components, a frequency offset of only a few MHz can result in a significant wind bias (1 m/s ≈ 3.75 MHz). To measure this effect and correct for the laser offset, 0.01% of the power laser light is diverted on the transmitter table into a fringe imaging etalon. By measuring the separation of fringe peaks between the seed light and power laser light, we record the frequency offset for each laser pulse. Details of the Laser Pulse Spectrometer (LPS) are described in a later publication (Fiedler et al., in preparation for Atmos. Meas. Tech.). Figure 3 shows a histogram of the offset for ~2.5 h of measurement on April 5, 2023. The distribution is nearly





symmetric, i.e. negative and positive deviations from the mean cancel each other in the wind processing algorithm (assuming constant pulse energy and a known atmospheric temperature). The frequency offset is measured to be taken into account in the processing chain.

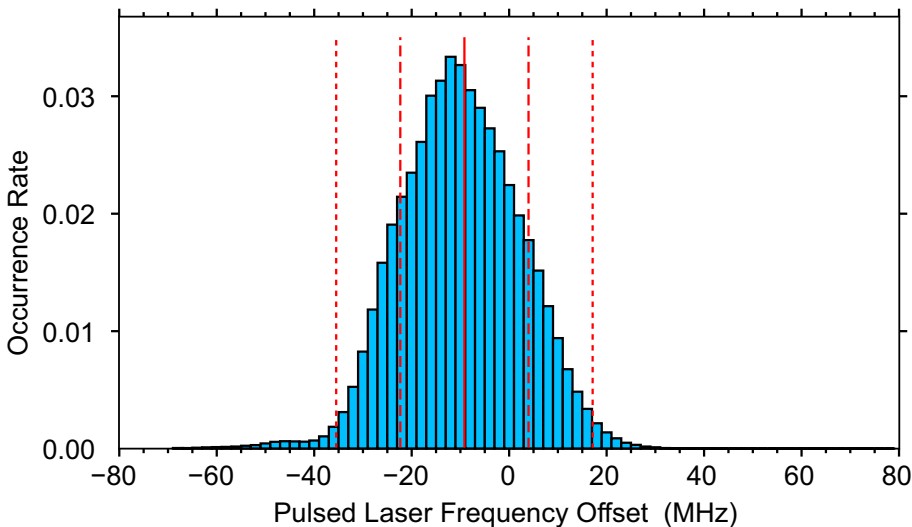

**Figure 3.** Histogram of frequency offsets between pulsed laser and seeder for ∼2.5 h (>900,000 laser pulses) of laser operation on April 5, 2023. Dashed and dotted lines show $1\sigma$ and $2\sigma$ limits, respectively.

## 2.2  Beam Pointing and Receiving Telescopes

The RMR wind lidar is set up as a bi-static system, with the laser being emitted off the optical axes of the telescopes. This
implies that we do not achieve overlap between the laser and the telescope's field-of-view (FOV) in the lowest part of the atmosphere. We aimed for a system where at least one telescope can be set vertically as well as tilted in different cardinal directions (e.g., north and west). In combination with a mono-static system, this would require complex beam steering optics. We, therefore, decided on the more cost-effective bi-static setup, taking into account that full overlap of laser and FOV is achieved above ∼30 km. In the following, we describe the beam-guiding, beam stabilisation and telescope setup in detail.
The two beams are directed by BGM1-A/B vertically up into the telescope room, where they are reflected by BGM2-A/B horizontally towards BGM3-A/B. Both BGM2 are motorized for automatic alignment of the beam if, e.g., the mirrors BGM3 are moved to another viewing direction of the lidar. BGM3-A/B direct the laser light into the atmosphere. We decided on a bi-static optical path, with the BGM3s mounted on poles beside the telescopes. The distance between the BGM3s and the optical axes of the tilted telescopes is about 1 m. BGM3-A/B are both equipped with motors for fine alignment of the beam
(see Appendix B for details). Path A can be directed either northward or westward at 65° elevation or higher, including zenith pointing. This requires steering the BGM3-A not only with the fine-alignment motors but additionally with a combination of a 360° motorized azimuthal mount and a ±10° goniometer. BGM3-B is fixed at 90° azimuth (eastward) and 65° elevation





because of mechanical restrictions by the building. This combination of optical mounts allows for a full range of motion and beam pointing, which can be controlled in real-time by software (see below).

The receiver assembly comprises two 70 cm Newtonian telescopes with motorized mounts allowing for tilting off-zenith and azimuthal rotation. We use laser-synchronized cameras mounted along the telescope's optical axis to ensure that the laser is correctly situated inside the telescope's field of view. For this, 10% of the received light is guided into the camera and 90% into the fibre for the detector. The basic principle of this setup is described in Eixmann et al. (2015) and used, e.g. in Gerding et al. (2016).

Monitoring the position of the laser beam in the telescope FOV allows us to stabilize our system alignment against thermal drifts, which can occur throughout the night, and it reduces the variability in signal strength by constantly optimizing the alignment. In practice, this is done by stabilizing the beam position every 5 s to the pixel position which corresponds to the best signal quality in each camera (Fig. 5 left).

The value of the stabilization system can be seen in Fig. 4. The left-hand panel shows a measure of lidar signal at 50 km

before the introduction of our beam stabilization system. We use black arrows to indicate the common slow misalignment that lidar systems can experience as the system thermally shifts throughout the night. These misalignments require an operator to occasionally optimise the signal by making a small adjustment to the beam pointing (marked yellow). The right-hand panel shows the same signal metric from a night when the beam stabilization system was active. The stabilization automatically compensates for beam drift and variability in the signal, with the result being a measurement with more consistent signal

quality throughout the night.

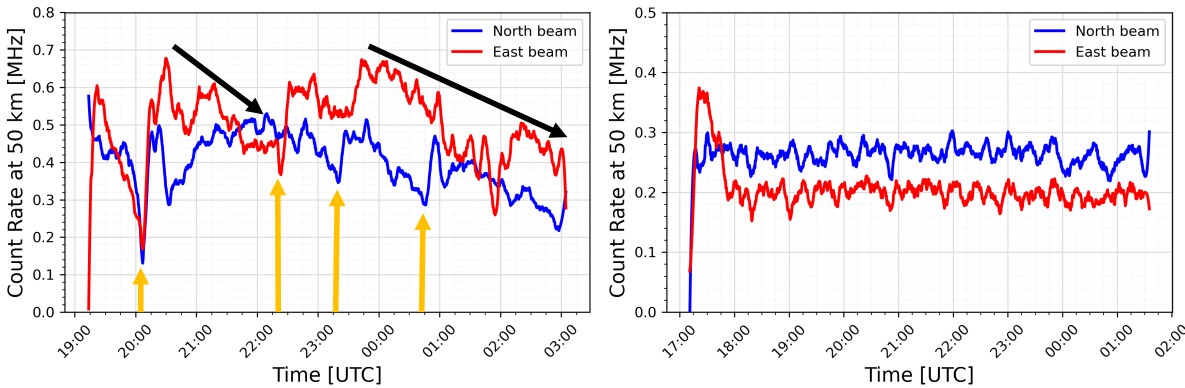

**Figure 4.** The effect of automatic beam stabilization using laser-synchronized cameras mounted on the telescope optical axis. The left-hand panel shows signals without beam stabilization, and the right-hand panel shows signals with stabilization. The blue line represents the lidar signal at 50 km in the North direction, and the red line represents the signal in the East direction. The black arrows represent slow misalignment, and the yellow arrows indicate times when the operator realigned the beam.

With the stabilization camera having a comparatively small FOV ($\sim 0.15°$ or 2.5 mrad), the laser may drift off the camera in between two soundings. We, therefore, attached another camera to the telescope, having $\sim 3.5°$ FOV for coarse alignment



of the laser beam (see Appendix C). Secondly, we can use this camera image of the laser beam against the background stars to determine the exact beam-pointing direction and elevation angle of our laser beam and telescope (Fig. 5 lower right). In practice, this is done by aligning the camera so that the laser tip is centred and uploading a still frame from the camera to an astrometry website (https://nova.astrometry.net). The website then gives the precise viewing direction based on the identified star constellation. The well-defined pointing information is crucial for avoiding systematic errors in the estimation of the zonal and meridional wind components.

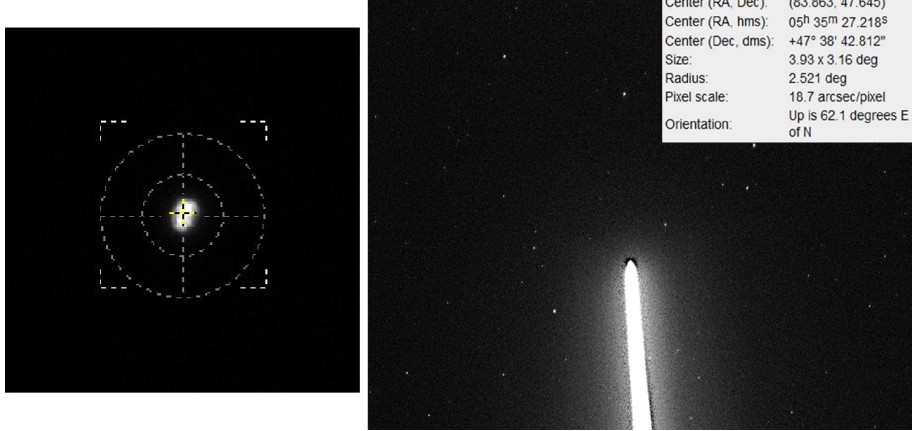

**Figure 5.** left: Cutout of an image from the gated small-FOV camera used for beam stabilization. The large dashed crosshair marks the optimal position, and the small yellow cross marks the calculated centre of the beam. right: Laser pointing check with the wide-FOV camera and star images. Results from astrometry calculations are shown in the inset (to be converted into ALT-AZ-coordinates).

## 2.3 Detection Bench Optics

The logical schematic for the detector bench is shown in Figure 6. Light enters the bench via three fibre optic cables, two fibres come from the telescopes (FIB-A and FIB-B), and one brings light from the seed laser (FIB-S) for use in determining the efficiency calibration between the detector channels. The lidar signals from the telescopes are 50 Hz interspersed so a second galvanometer scanner mirror, similar to the one on the transmitter bench and using the same 50 Hz trigger, is used to direct the pulses down the correct optical path. Both beams are collimated (COL), and then the dichroic mirror (D-608) separates the Nitrogen Raman signal (608 nm). The Raman signal is sent through a 608 nm interference filter (IF-608) and is focused onto a single photon counting module (VVR). The Doppler-Rayleigh signal passes through a 532 nm interference filter (IF-532) and is split with a 30:70 beam splitter at (BS-1). The reflected 30 % is sent to a 95:5 beam splitter (BS-2) where it is transmitted onto the high gain detector channel (VH). The 5% reflected light is focused onto the low gain detector channel (VL). The remaining 70 % of the light at BS-1 is directed through the iodine vapour cell (IOD) and is split by another 95:5 beam splitter (BS-2) before being directed to a high gain channel (VDH) and a low gain channel (VDL). All these detection channels are equipped with Avalanche Photo Diodes (APD).



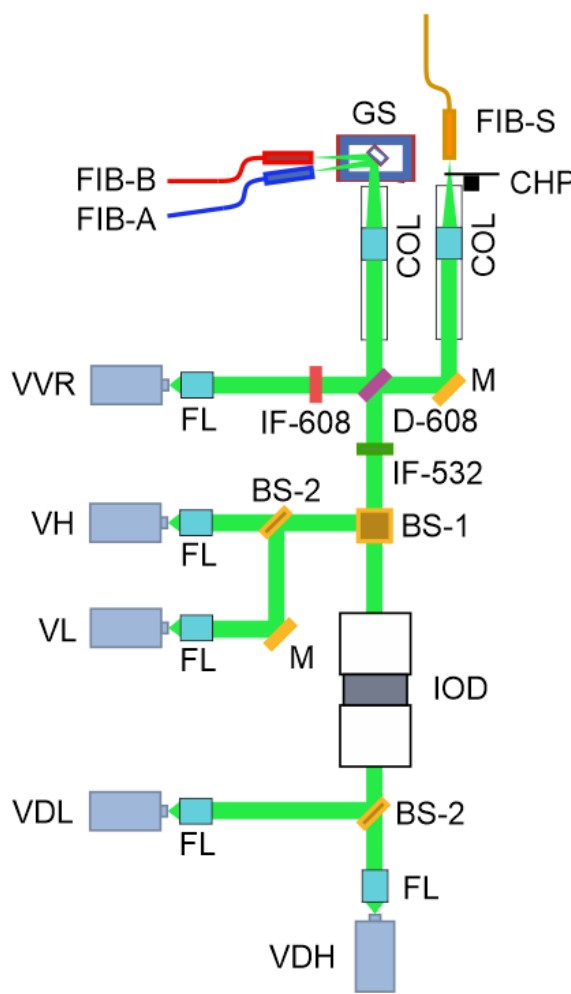

**Figure 6.** Schematic of the detection bench. FIB-A/FIB-B/FIB-S: fibres from telescopes A or B or the seeder; GS: galvanometer scanner; COL: collimator; CHP: chopper; D: dichroic mirror; M: mirror; IF: interference filter; BS: beam splitter; FL: focussing lens; IOD: iodine cell; VVR, VH, VL, VDH, VDL: detectors of Raman channel, high/low Rayleigh channel, high/low Doppler channel.

The detected signals are recorded by an FPGA-based counting system with dedicated pulse counting software for up to 16 channels. The FPGA clock rate of 100 MHz allows for a physical range resolution of 1.5 m. Up to 3 photons per bin can be recorded. An IAP-built signal-conditioning board is used to adapt the signal level of the APD pulses to the FPGA input level 270 and to discriminate electronic noise. Backscatter profiles for each laser pulse and each detection channel are transmitted to a PC via a USB-3 connection. The PC runs a Python code for real-time data acquisition and storage. The overall system is called LISA, LIdar Single pulse Acquisition, and runs automatically without operator action. All operator-based actions are performed with external software if needed for, e.g., alignment purposes (see Section 2.4).



Besides the backscattered light of the pulsed laser, we feed the seed light via the fibre FIB-S (see Fig. 6) into the detector to
get an independent calibration metric for the system (Hildebrand, 2014). The seed light follows nearly the same optical path
as the lidar signal onto each APD. It is passed through a mechanical chopper (CHP, Thorlabs MC2000B) so that it appears in
the detector only at a range of approximately 300 km and above. The light passes through some collimating optics (COL) and
is reflected by a mirror through and off of the backside of a dichroic filter (D-608).

Knowledge about the exact iodine absorption spectrum is crucial for the wind calculation from the VDH/VH ratio (or
VD/VL for the low channels). Figure 7 shows the modelled absorption spectrum for gaseous molecular iodine at 40 °C (blue)
produced using IodineSpec software from the Institute of Quantum Optics at Leibniz University in Hannover/Germany. The $I_2$
line 1109 has been selected for optimal wind sensitivity within the wavelength range that is covered by the Nd:YAG laser. The
measured spectra are shown as magenta (SCS, Seeder Cell Spectrometer) and green (detector) lines in Fig. 7. The SCS cell
serves only to stabilise the seeder, with the stabilization point marked by the black star. By construction, the laser frequency
is optimally fitting the absorption cell in the detection bench. The transmission of the cell depends on the Doppler shift of the
backscattered light and therefore a direct measure of the wind speed (Baumgarten, 2010). The red line in Fig. 7 shows the solar
spectrum. The selected $I_2$ line is close to the centre of a weak Fraunhofer line, which reduces the background by approximately
30% for the vertical pointing, daylight-capable temperature lidar and planned future daylight wind measurements.

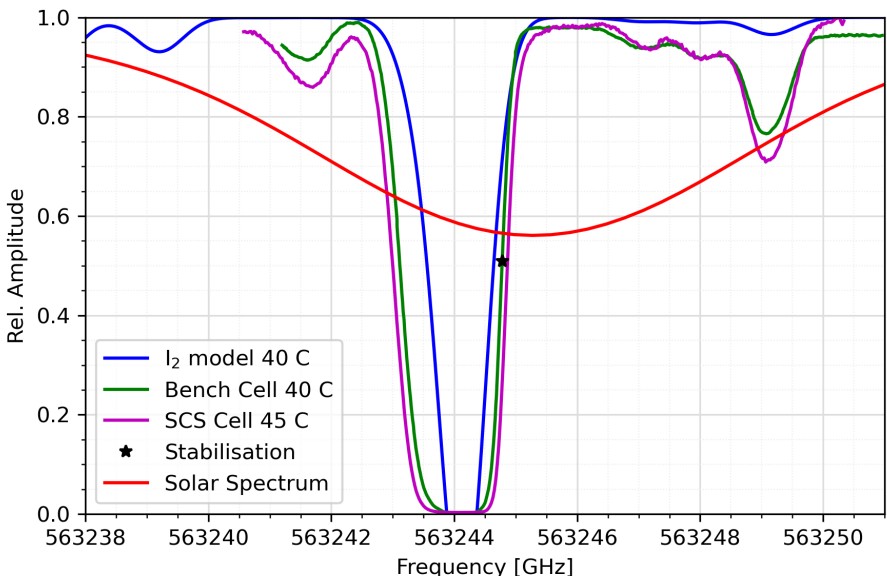

**Figure 7.** Plot of relative absorption vs. frequency for the molecular iodine gas cells used as atomic references in this system. Blue: $I_2$ model
of Leibniz University Hannover, green: detection bench cell, magenta: SCS cell, red: solar spectrum (J. Aboudarham, 2020)

.





## 2.4 Kühlungsborn Lidar Automation Software (KLAUS)

Our wind lidar is designed as a modular system with independent units for triggering, laser control, beam steering, safety etc. The subunits share different labs and are connected to the local network. All subunits are connected to a software back-end that is responsible for the overall control of the system. All operator functions are bundled in a single front-end, allowing the operation of the lidar also with less experienced personnel.

The Message Queuing Telemetry Transport (MQTT) protocol forms the basis for our lidar systems networking concept. All
of our computers, lasers, optoelectronics, triggers, safety systems, interlocks, detectors, data acquisition, and control software are networked by "subscribe and publish" under a common framework. This system has proven efficient and useful for coordinating multiple hardware devices (with disparate proprietary software elements) into a single, cohesive lidar system. The status messages are typically broadcasted with $0.5 - 1\,\mathrm{Hz}$ cadence. This is fast enough for the overall control of the lidar, even if the different devices work and communicate on a much higher cadence (up to $100\,\mathrm{Hz}$).

The Kühlungsborn Lidar Automation Software (KLAUS) is the heart (i.e. back-end) of the lidar operation software. It is a multi-thread state machine coded in Python with two main sections. One section (the actual state machine) is doing the sequential startup and shutdown of the different subunits. The states are connected by actions like, e.g., powering on the different hardware parts, activating the laser safety devices, starting the laser, or initializing the beam steering. The action requests are broadcasted to the subunits by MQTT messages, and the successful execution is messaged the same way, setting
the system to the next state. Fig. 8 describes the KLAUS concept in more detail.

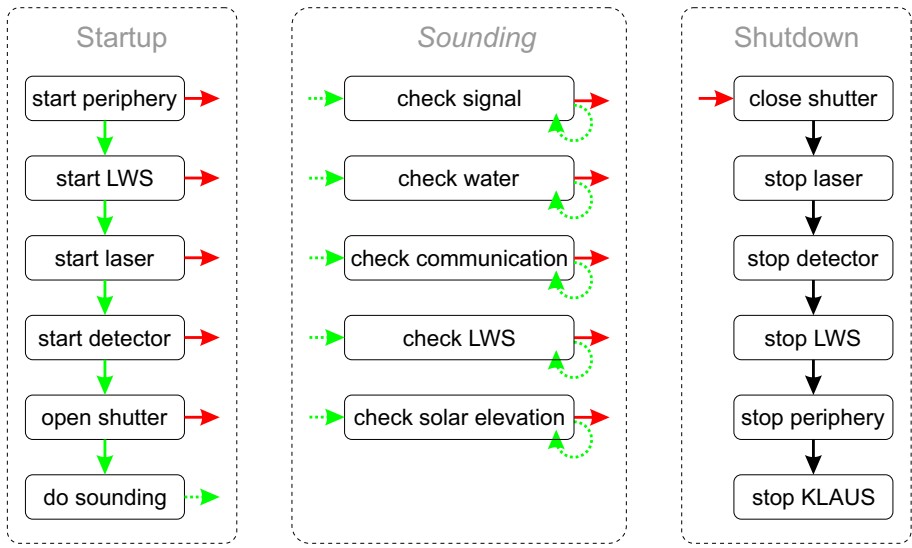

**Figure 8.** Schematic of the KLAUS actions that connect the particular states of the system. Only the most important actions are shown for clarity. Green arrows show the flow in case of success, and red arrows in case of problems with a particular action. In the shutdown process, the success is monitored, but in either case, the next action is triggered (black arrows). The dotted arrows show the threads inside the 'Sounding' state.



The system reaches its final startup state by opening the laser shutter. The sounding i.e. data acquisition starts automatically as soon as the trigger signal from the power laser is available. The two laser beams are steered by the beam stabilization software to their respective targets on the cameras attached to the telescopes once the backscattered light has been detected by the camera (see Fig. 5 left) . The sounding state contains a check of the most important system parameters, which is the second main section of the script. If needed, the KLAUS script shuts down the whole lidar automatically. The system check comprises different countdown timers that are put on hold as long as the particular criterion is fulfilled. If not, the timer counts down until either the criterion is met again or the time has expired. The former resets the timer, the latter initiates an automated complete shutdown of the lidar. Checked criteria include the actual and 30-minute mean signal level, the actuality of the data, actions of the laser safety system (see below), watchdog messages of all subsystems, laser conditions and cooling water flow. For example, the lidar is shut down in the morning twilight, if the signal is too low for 30 min, or if the secondary laser cooling water does not flow for 10 s. The operator can always take over and override the automated systems checks.

Some critical parts of the lidar receive redundant control by the particular sub-software, independent of the KLAUS script. For example, a rain detector which is connected to the hatch control sends a direct signal to close the hatch in case of rain. The laser firmware stops the laser automatically if the primary cooling water flow or laser head temperatures are out of limits. The KLAUS script reacts to such events with a countdown of, e.g., the signal timer and initiates the shutdown of the whole lidar after a defined period. The operator is always notified about the state of the systems and the shutdown, mainly because of safety concerns: a Class IV laser operating in a shared lab/office building and propagating into unrestricted airspace. Notifications about shutdown ("regular" because of persistent cloud coverage or "irregular" because of technical problems) are automatically sent to the operator by a Telegram Messenger bot included in the KLAUS Python script. Further redundant safety measures cover some interlock software that shuts down the power laser, closes the hatch of the telescope room, and notifies the operator automatically in case of a lost connection to the KLAUS script (watchdog).

While KLAUS is a flexible Python script for lidar control, it does not comprise a graphical user interface (GUI). KLAUS solely receives and sends information via either MQTT or, in special cases, TCPIP socket connections. The GUI is developed with the open-source development tool Node-RED. This browser-based tool can run on any PC connected to the same MQTT broker but is for practical reasons installed on the same PC as KLAUS. The GUI offers a comfortable interface between the human operator and the KLAUS automation software (see Figure 9). The Node-RED front end receives status information as well as text messages and displays them in the GUI. Different buttons in the browser display send commands for, e.g. stepwise or complete, start or shutdown of the lidar. All information between KLAUS and the Node-RED front end is exchanged via the MQTT broker. The GUI additionally displays information about the lidar signal quality, that is received via MQTT from the LISA counting system. Furthermore, different webcam photos provide good visual feedback to the operator. Overall, the KLAUS script allows either a manual, GUI-guided operation or a semi-automated operation, where the script takes over control after starting the lidar manually.





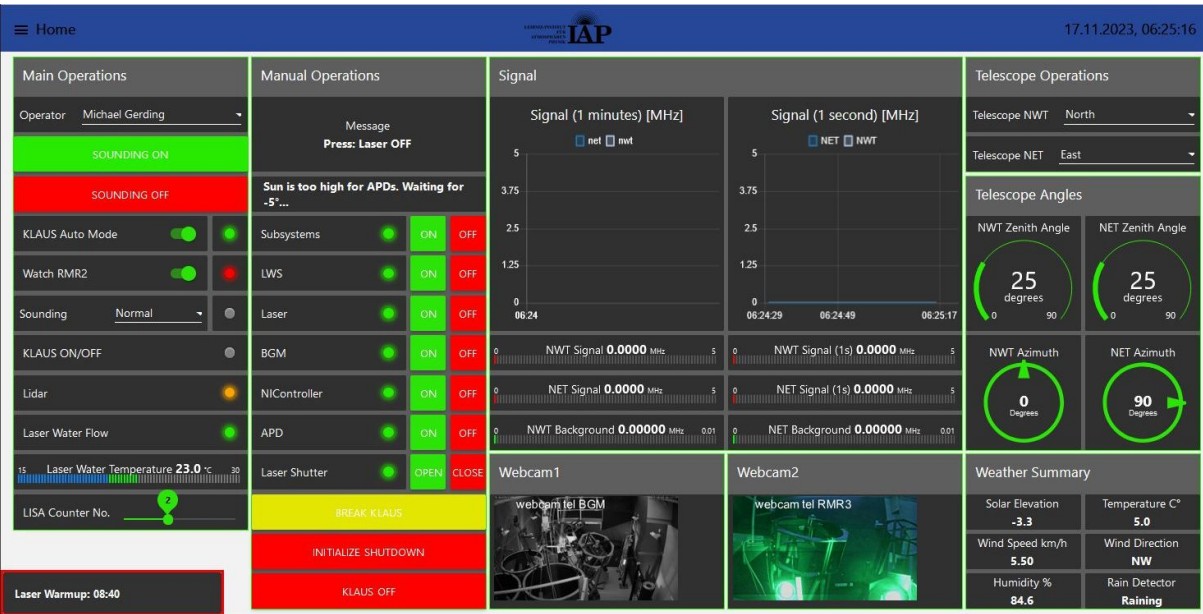

**Figure 9.** Screenshot of the graphical user interface of the RMR lidar.

Some other sub-systems are worth describing that belong to the lidar operation package. As mentioned above, these make use of different coding languages, operating systems (Linux and Microsoft Windows) and hardware platforms (e.g., Raspberry Pi computers). A so-called Laser Warning System (LWS) informs the personnel in the lab/office building about the laser operation via displays on all doors with laser access. At the same time, these doors are watched by the LWS and the laser shutter is automatically closed for safety reasons if one of the doors is opened. A receiver for Automatic Dependent Surveillance-Broadcast (ADS-B) aeroplane transponder signals is included in the LWS. The laser shutter is closed if an aircraft gets close to the beam in the sky, and opened again afterwards. Because ADS-B is not mandatory for aircraft, four modified Furuno M1835 radars watch the airspace above the institute in different directions. In the case of air traffic within an angle of $\sim20°$ around the laser beams, the laser shutter is automatically closed. Besides these systems for laser safety, we installed some hardware for housekeeping purposes (monitoring of temperatures of air, water, laser housing, and cooling water flow). The electric roof was installed in 1996 without a computer interface. We developed a computer interface for remote operation to include it in our KLAUS concept. As mentioned above, a rain sensor acts directly on the hatch controller and triggers the hatch to close in case of precipitation.

The daylight-capable temperature lidar (RMR2) has used a previous version of the KLAUS software since the autumn of 2019. Recently in May 2023, we updated the RMR2 lidar with a new laser, laser controller, APD controller etc. Both lidars have now about the same technical level. Therefore, we can now operate RMR2 with the same KLAUS version as RMR3. Both lidars can be operated in parallel or independently.



## 3 Data Processing and Data Products

In this section we will briefly describe the data processing procedure, describe the standard Level 1 and Level 2 lidar data products, provide specific case study examples which illustrate typical lidar measurements, and discuss data quality assurance and availability. A complete description of the algorithm design and data treatment will be given in a future companion article.

While the lidar is in operation, the photon count data files from the lidar and housekeeping files from individual instruments in each sub-system described in the previous sections are written, transferred, and stored in various file formats. This raw data represents "Level 0" lidar data and is stored in a heterogeneous fashion across various computers and servers. We have an automatic script which operates once per day that reads these lidar files and concatenates them into a single daily file on a common time grid. During this concatenation procedure, poor quality data is masked, some basic lidar corrections are made to the data (range, background, deadtime etc.) and ancillary data (telescope angle, laser frequency offset, etc.) is collected and matched with corresponding lidar photon count data. This concatenated, cleaned, and corrected file is referred to as "Level 1" and represents data which is ready to be used for determining geophysical variables.

To transform "Level 1" files into scientific data products, a second batch code is used to ensure that all data is processed uniformly. We derive density, pressure, temperature, doppler ratio, meridional wind, zonal wind, stratospheric aerosol backscatter and noctilucent cloud backscatter as a function of geometric altitude and time. These physical variables are retrieved at five standard filter resolutions: 5 minutes by 41 metres (representing the smallest-scale features of interest near the Brunt–Väisälä frequency), 15 minutes by 500 metres (for high-resolution studies), 30 minutes by 1000 metres (to capture features of mid-scale gravity waves), 90 minutes by 1500 metres (to push the measurements higher into the mesosphere), 120 minutes by 3000 metres (to make nightly average profiles). These filtered geophysical variables represent "Level 2" data and are ready for use.

In the single-edge Doppler-Rayleigh technique, the density, pressure and temperature are derived using the classical method of hydrostatic integration of Hauchecorne and Chanin (1980) and comply with the Network for the Detection of Atmospheric Composition Change (NDACC) standards for vertical resolutions and uncertainty reporting for Rayleigh lidars (Leblanc et al., 2016a, b). Stratospheric aerosol measurements are made by comparing the Rayleigh signal at 532 nm to the vibrational Raman signal at 607 nm, as is described in Alpers et al. (2004). The noctilucent cloud backscatter measurements are made following the procedures established in Gerding et al. (2013). Doppler ratio and wind component measurements are made following the technique established in Baumgarten (2010). A more complete description of scientific data products will be made in a future companion article: The Doppler wind, temperature, and aerosol RMR lidar system at Kühlungsborn/Germany – Part 2: algorithm design and error propagation.

In Fig. 10 we show an example of a night of coincident lidar wind and temperature measurements from the 6th of February 2023. System A was directed 25° to the North and System B 25° to the East. In the left-hand column, we see the time-resolved contour plots of temperature (top), zonal wind (middle), and meridional wind (bottom). In the time-resolved zonal winds, positive values (red) represent eastward flow, and in the time-resolved meridional winds, positive values (red) represent northward flow. In the right-hand column, we see the corresponding nightly average profiles. The average temperature is shown in green (top), the average zonal wind in red (middle) and the average meridional wind in blue (bottom). Overplotted in black is



the nightly average ECMWF-IFS profile during the measurement time. In all cases, the shaded region is the standard deviation
of the nightly measurement and serves as an indication of the natural geophysical variability over the measurement period.
We can see that there is better agreement between the lidar measurement and ECMWF in the stratosphere. In the mesosphere,
above 50 km, ECMWF is in poor agreement with observations and lacks many of the oscillations associated with the observed
gravity waves. Additionally, the standard deviation of the ECMWF profile is much smaller than the observations, indicating
that natural variations and waves are not captured completely in the ECWMF data.

An interesting feature present during this night is a Mesospheric Inversion Layer (MIL) near 75 km in the temperature data
(top left). In the time-resolved panels for temperature, we can see that the amplitude of the layer began to increase around 01:00
UTC and reached a peak around 04:30 UTC. At the corresponding times and altitudes, the zonal wind (middle left) shows the
zonal wind reversal coupled to the MIL. The assumed understanding is that wave packets are breaking on the mean winds at
the height of the MIL, depositing momentum which acts to slow and reverse the flow. The resultant thermal energy which is
released sustains and amplifies the thermal MIL and acts to modify the local wave stability criteria in positive feedback, which
supports continued wave filtering.



**Figure 10.** Coincident wind and temperature profiles on the night of the 6th of February 2023 measured by the Kühlungsborn lidar. The left-hand column contains the time-resolved panels of temperature (top), zonal wind (middle) and meridional wind (bottom). For the winds, northward and eastward flow are positive (red) and southward and westward flow are negative (blue). The right-hand column contains the nightly average lidar profiles along with the average ECMWF profiles for temperature (top), zonal wind (middle) and meridional wind (bottom). The shaded regions represent the standard deviation of the nightly average and serve as an estimate of geophysical variability. The resolution of the data in this figure is 2 hours by 3 km.





## 4 Summary

The upgraded Doppler RMR lidar at Kühlungsborn went into operation in October 2021. It adds capabilities for measuring horizontal wind in the middle atmosphere up to $\sim$90 km altitude during nighttime without interrupting the decadal scale soundings

with the vertically viewing daylight-capable RMR temperature lidar. The upgrade makes use of the single-edge iodine-filter technique for the detection of the wind-induced Doppler shift of backscattered light. This technique allows measuring simultaneous and co-incident temperatures along with the winds by the same tilted beams. The upgraded RMR lidar makes use of an improved version of our lidar operation software (KLAUS: Kühlungsborn Lidar Automation Software). KLAUS allows semi-automated operation of the lidar, i.e. it can run autonomously including systems checks and automated shutdown for clouds

after the lidar is started manually by an operator. Also, the startup process is to a large extent simplified and automated.

The paper describes all relevant details for the design and construction of a Doppler RMR lidar using the single-edge technique. The state-of-the-art diode-pumped Nd:YAG laser is externally seeded by a laser whose wavelength is locked to an iodine absorption line. The laser serves both viewing directions simultaneously with 50 Hz each. While beam-guiding and receiving telescopes are duplicated, the detection bench consists of one single chain of optics. By this, we avoid systematic

differences between the results in different viewing directions. Motorized components allow for complete software control of the lidar. The beams can be steered in different viewing directions and are automatically stabilized to the telescopes' FOV for the whole time of the sounding. Furthermore, our control and surveillance mechanisms cover a suite of components like (i) a laser pulse spectrometer (LPS) for measuring the frequency offset of the pulsed laser, (ii) redundant air-space surveillance with automated blocking of the laser in case of air traffic, (iii) redundant housekeeping and weather monitoring with the automated

shutdown of the systems if technically needed.

The overall design of the lidar allows for extensive soundings whenever sky conditions are favourable. The automation concept strongly reduces the necessary manpower as well as the necessary qualification of the personnel. To date, we acquired 944 hours of data over 159 nights since October 2021. We have presented an example of time-resolved wind measurements covering the middle/upper stratosphere and the mesosphere up to $\sim$90 km. The data show typical variations in the meridional

and zonal winds of $\pm$40 m/s in the middle mesosphere, increasing above. Observed wind variability in the mesosphere is much larger than calculated by, e.g., ECMWF-IFS. Our retrieval code is standardized and automated, assuring continuous and high data quality. The data processing chain will be described in a companion paper.

*Code availability.* CAD drawings and STL files for 3d printing of adapters used for mounting the 150 mm beam guiding mirrors are available at https://igit.iap-kborn.de/gerding/rmr3-technical-description.

*Data availability.* Lidar data in this paper is available at https://www.radar-service.eu/radar/en/dataset/jqPubzXuXQJzoeBb



**Table A1.** Laser bench component specifications, selected items (cf. Fig. 2)

| label | component | specifications |
|---|---|---|
| laser | | Innolas Spitlight DPSS EVO IV |
| | | rep. rate, pulse length: 100 Hz, 8 ns |
| | | pulse energy (532 nm): ∼500 mJ |
| seed laser | | Coherent Prometheus 100 |
| | | stabilized to I2 line 1109 (563244.8 GHz) |
| | | wavelength: 532.112 nm (air), 532.260 nm (vac.) |
| BP | beam picker | double wedge plate |
| | | fused-silica, AR-coated |
| | | Laseroptik GmbH |
| CHP | chopper | Thorlabs MC2000B-EC with plate MC1F2 |
| BS | beam splitter | Thorlabs CCM1-BS013/M |
| GS | galvanometer scanner | Scanlab dynAXIS 3S |
| | | OEM amplifier: mini-SSV |
| | | offset drift: <5 $\mu$rad/K |
| | | repeatability: <1 $\mu$rad |
| | | mirror: 15.9 mm x 21.0 mm (width x height), fused silica |
| | | 532 nm coating: Laseroptik GmbH |
| | | reflectivity: >98.5% @ 41° − 49° incidence |
| mirror | | Laseroptik GmbH, 532 nm |
| | | 25 mm dia. x 9.5 mm  fused silica |
| | | reflectivity: >98.5% @ 41° − 49° incidence |
| PD | photo diode | Thorlabs DET10A/M |
| BWT | beam widening telescope | Sill Optics, 10x |
| | | wavelength: 532 nm |
| | | entrance beam diameter: <10 mm |
| BGM1-A/B | beam guiding mirror | Laser Components GmbH, 532 nm |
| | | 150 mm dia. x 20 mm, fused silica |
| | | reflectivity: >99.5% @ 30° − 60° incidence |
| | | mount: Thorlabs KS3 with adapter (IAP) |

## Appendix A:  Laser Bench Optoelectronics

Details of the laser bench elements are given in Table A1.





**Table B1.** Beam guiding specifications, selected items

| component | specifications |
|---|---|
| BGM2-A/B | mirror: see BGM1-A/B in Appendix A |
| | mount: Thorlabs KS4 with IAP adapter (facing downward) |
| | motor: Thorlabs Z812 with KDC101 controller |
| BGM3-A/B | mirror: see BGM1-A/B in Appendix A |
| | mount: OWIS TRANS 100 (customized) with IAP adapter |
| | motor: Newport TRA25PPD with SMC100PP controller |
| | 25 mm travel range, 60 N push force, $\pm 0.18\,\mu$m repeatability |
| BGM3-A (additional mount) | goniometer: OWIS MOGO 150 with PS90+ controller |
| | $\pm 10°$ travel range, <0.01° repeatability |
| | rotation stage: OWIS DMT 130N with above PS90+ controller |
| | 360° travel range, <0.01° repeatability |

## Appendix B: Beam guiding mechanic

Details of the beam guiding mechanics are given in Table B1.

## Appendix C: Telescope and Detection Bench Optoelectronics

Details of the receiving telescopes and the detection bench are given in Table C1.

## Appendix D: Timing and Trigger Control

The three beams, double power laser setup of our Doppler-Rayleigh-Mie-Raman lidar presents a technical challenge. We coordinate the off-zenith beams which share one power laser and the zenith pointing beam which has a separate laser. Each laser pulse must be correctly timed and measured, with the appropriate signal delays introduced between the two lasers to avoid signal contamination. Additionally, each laser must be frequency controlled and assessed on a pulse-by-pulse basis by the laser pulse spectrometer LPS (cf. Sect. 2.1). Central timing is done with three National Instruments sbRIO-9637 Single Board Controllers (see TrigDist and two LasCtrl in Fig. 1 and Fig. D1). Controller software as well as the graphical user interface are coded in LabView. The flexible, modular design of the Trigger Controller allows for all trigger signals to be software-controlled and allows the operator to set individual delays and pulse lengths for both normal operations and specific experimental requirements.

Figure D1 presents a logical schematic for the propagation of trigger pulses through the lidar system. The Trigger Distributor (TrigDist) generates a 100 Hz main trigger signal which is guided to the Laser Controllers (LasCtrl-2 and LasCtrl-3) for



**Table C1.** Receiver specifications, selected items (cf. partly Fig. 6)

| label | component | specification |
|---|---|---|
| telescopes | | Astelco Systems, 70 cm diameter |
| | | Newton design with motorized Alt-Az mount |
| | | focal length: ~265 cm (F/3.8) |
| | | cage tube: carbon-fibre rods |
| beam guiding cameras | | Basler acA640-120gm (2 for each system) |
| | | 120 fps, 659 x 494 pixels, monochrome |
| | | lens for small FOV: Schneider Xenoplan 1.4/17-0903 (f = 17 mm), through telescope |
| | | lens for wide FOV: Navitar NMV-50M1 (f = 50 mm) |
| FIB-A/B/S | optical fibre | Ceram Optec |
| | | core diameter 300 $\mu$m, NA 0.13 |
| GS | galvanometer scanner | Scanlab dynAXIS 3S |
| | | OEM amplifier: SSV30 |
| | | offset drift: <5 $\mu$rad/K |
| | | repeatability: <1 $\mu$rad |
| | | mirror: 18.4 mm x 10.7 mm (width x height), silver-coated silicium |
| CHP | chopper | Thorlabs MC2000B-EC with plate MC1F2 |
| D-608 | dichroic mirror | Laseroptik GmbH |
| | | HR 608 nm, HT 532 nm |
| IF-608 | interference filter | Barr Associates (now: Materion Corporation) |
| | | central wavelength: 607.5 nm |
| | | spectral bandwidth: 0.34 nm, 2 cavities |
| | | max. transmission: 87% |
| IF-532 | interference filter | Barr Associates (now: Materion Corporation) |
| | | central wavelength: 532.1 nm |
| | | spectral bandwidth: 0.13 nm, 2 cavities |
| | | max. transmission: 77% |
| BS-1 | beamsplitter cube | Thorlabs BS019 |
| | | R/T: 30/70 |
| BS-2 | beamsplitter plate | Thorlabs BSF20-A |
| | | reflectivity: ~5% |

triggering the Nd:YAG lasers, and used on the detection bench. Both the new off-zenith beams and the original vertical pointing
lidar are synchronised to this main trigger. Each LasCtrl also controls the buildup-time (BUT) of the Nd:YAG to ensure optimal





**Table C2.** Receiver specifications, selected items (contd.)

| label | component | specification |
|---|---|---|
| VH | photon counting channels | Avalanche Photo Diodes (Excelitas Technologies) |
| VL | | type: SPCM AQRH-16 |
| VDL | | sensitive area: 0.17 mm |
| VDH | | quantum efficiency: >50% @532 nm, >65% @ 608 nm |
| VRR | | dark counts: <25 Hz |
| | | power supply: IAP |
| | | VH/VL: High/Low Rayleigh reference channel |
| | | VDH/VDL: High/Low Rayleigh Doppler channel |
| | | VVR: $N_2$ vibrational Raman channel (608 nm) |
| counter | | FPGA module CESYS EFM-02/B150-3I |
| | | FPGA: Xilinx SPARTAN-6 |
| | | clockrate: 100 MHz (bin width: 1.5 m) |
| | | channels: 16 |
| | | signal conditioning: IAP |

laser seeding and laser frequency stability. For testing purposes, the LasCtrl can also generate the laser trigger pulse internally. Further details regarding laser buildup-time reduction (BUTR) and laser frequency stability can be found in section 2.1.

After laser emission and beam separation into beam paths A and B, the laser pulse is recorded by photodiodes, whose signals are again received by the TrigDist. Here they are used to trigger the beam stabilisation cameras, synchronise the gating of the APD detectors, and start the photon counters (labelled C, APDCtrl and LISA in Fig. 1). Finally, the original 100 Hz signal is used to synchronise a chopper in the detection bench as well as the galvanometer scanners on the laser bench and the detection bench. The analogue drivers for the galvanometer scanners are additionally included in the LasCtrl and TrigDistr hardware (see Sect. 2.1 and 2.3).

Information about the direction of either pulse (system A or B) has to be kept throughout the triggering system to allow proper integration in the two beam directions by the photon counting system, LISA. Both sub-systems are again able to switch between the 2-beam mode and the sounding in only one direction.





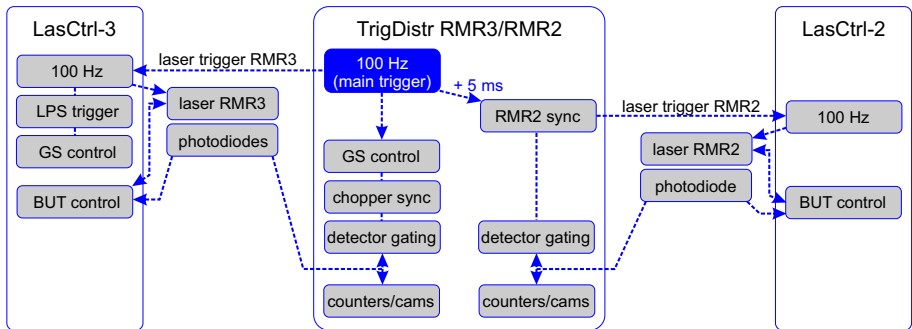

**Figure D1.** Logical schematic for the trigger distribution system.

*Author contributions.* MG and GB secured the grant funding and led the project. All co-authors worked on system design, and construction, and participated in observations. MG and RW wrote the manuscript with the help of all co-authors.

*Competing interests.* Robin Wing and Gerd Baumgarten are members of the editorial board of Atmospheric Measurement Techniques.

*Acknowledgements.* This project is funded under the title "Analyzing the Motion of the Middle Atmosphere Using Nighttime RMR-lidar Observations at the Midlatitude Station Kühlungsborn (AMUN)" by the Deutsche Forschungsgemeinschaft (DFG) - Projektnummer 445400792. The lidar data from this project are connected to project W1 (Gravity Wave Parameterization for the Atmosphere) of the Collaborative Research Centre TRR 181 "Energy Transfers in Atmosphere and Ocean" funded by the Deutsche Forschungsgemeinschaft (DFG, German Research Foundation) - Projektnummer 274762653. We thank Franz-Josef Lübken for his stimulating ideas that substantially supported the
initial phase of this project. We would also like to acknowledge the invaluable technical support from Mr. M. Priester and the support during observations from various enthusiastic summer students.



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
