# Peer review of "The Doppler wind, temperature, and aerosol RMR lidar system at Kühlungsborn/Germany – Part 1: technical specifications and capabilities"

_EGUsphere, 2023_

## Author Comment (AC1)

We thank the reviewer for the careful work. We are happy that he/she agrees to our concept of providing many technical details. In the following, we reply to the individual comments (in blue).

This manuscript described the Rayleigh-Mie-Raman (RMR) lidar system at the The Leibniz Institute of Atmospheric Physics (IAP) in Kuhlungsborn, Germany that is one of research groups having much of the lidar measurement know-how.

As mentioned in the Introduction, the importance of lidar measurements, which can observe temperature and wind speed in the middle atmosphere with high time and height resolution, is widely accepted. However, there are not many sites in the world that have such lidar facilities because of the complexity of the system configuration and operation. This paper presents the design concept of the RMR lidar system, an overview and detailed description of each component, and the arrangement of the optical elements. Each component itself may not necessarily be new technology, but the detailed information of where, why, and how it was incorporated into the lidar system is very important when a lidar system is built. This paper is a valuable insight for the lidar research community as well as for newcomers to it. The data analysis will be described in a companion paper, so there is no need to go into detail in this paper, but it would be nice to have more information on data quality for the example observations. So, I would recommend it for acceptance after the minor points listed below are addressed.

(Minor comments)

- In each section, abstract, summary and others, it is better to use same words for your lidar system.
    - a vertically emitting, daylight-capable temperature lidar (called 'RMR2' here)
    - a two-beam tiltable system intended for wind and temperature measurements (called 'RMR3' here)
    - "3-beam Doppler-Rayleigh wind lidar system" and "vertically pointing daylight-capable Rayleigh-Mie-Raman (RMR) temperature lidar with a 2-beam, nighttime only RMR wind-temperature lidar" are used in abstract.
    - "Doppler RMR lidar" is used in summary.

    We apologize for confusion with the naming of our lidar systems. We will unify the wording and use the term "RMR temperature lidar (RMR-T)" for the older system and "RMR wind-temperature lidar (RMR-WT)" for the new system that is described here. Terms *RMR2* and *RMR3* will be replaced by the more descriptive *RMR-T* and *RMR-WT* in text and figures. Number of beams and daylight-capabilities will only be mentioned if useful. The term "Doppler RMR lidar" will still be used wherever this general type of lidar is meant.

- (Line 35) Check "between 30 and 80 km" and remove "(?)".

    We will correct the reference to Rüfenacht et al., AMT, 2012.

- Add power consumption of laser in Table A1.

    We will add "power supply: 3 kW".

- (Fig.10) I would like to recommend you that a typical data is shown as an example for this paper because Discussion of the observed phenomena is not main purport. The night on 6 Feb. 2023 might not be a good example because it was between minor warming and major warming.

  The situation in Fig. 10 is typical for winter conditions at Kühlungsborn, where the lidar is often located at the edge of the polar vortex and the spatial/temporal variability in the middle atmosphere is large. Therefore, from our point of view, this example demonstrates the need for localized measurements. We will add the following sentences at the end of the description of Fig. 10 in order to set the figure into context: *"We expect better agreement between ECMWF output and observations in the summer, when the variability in the middle atmosphere is much smaller. Nevertheless, this typical winter example demonstrates the need for local measurements of winds and temperatures for understanding of dynamics in the stratosphere and mesosphere."*

- (Line 389-390) Mention the measurement errors of temperature and wind speeds, too. Comparison of error and standard deviation is necessary when the natural geophysical variability over the measurement period is discussed.

  We will add the following sentences: *"The statistical uncertainty of the temperature profile depends on the photon count rate and is omitted here for clarity of the figure. The uncertainty of the nightly mean temperature profile is ~0.2% at 40 km and 4 % at 70 km (41 m resolution). Calculations of wind uncertainties have to include not only the photon statistics, but also the gradients in the calibration matrix at the particular wind speed and temperature, and the spectral distribution of laser pulses (cf. Hildebrand, 2014). As a rough estimate, we get ~0.7 m/s at 40 km and 6 m/s at 70 km altitude (nightly mean, 41 m resolution). A detailed error description will be provided in the companion paper."*

- (Line 423) "~since October 2021", to when? If it is November 2023, "159 nights" are approximately 20% of 26 months. Is the weather condition only reason of no observation in 80% of nights?

  We had some technical issues in the first winter, but the observations are mainly limited by weather conditions since spring 2022. In the revised version, we will state that the number of nights is calculated until **.

We like to note that we will clarify the filtering method in the description of Fig. 10. A Gaussian filter is applied with the numbers describing the full width at half maximum. This applies also to the last sentence in the caption of Fig. 10.

---

## Author Comment (AC2)

*We thank the reviewer for their feedback! We reply to the comments below (in blue color). Line numbers refer to the original submission.*

This paper effectively elucidates the Rayleigh-Mie-Raman (RMR) lidar system deployed at the Leibniz Institute of Atmospheric Physics in Germany. It underscores the significance of lidar measurements in the middle atmosphere and emphasizes the global rarity of such facilities due to their intricate nature. The author provides a comprehensive breakdown of the RMR lidar system, offering a lucid overview of each component and their interconnections.

While the individual components may not introduce entirely novel technologies, the paper stands out in its emphasis on explaining the rationale behind their integration into the lidar system. This aspect proves valuable for both seasoned researchers and newcomers seeking a deeper understanding of the system's operational principles.

*We are pleased that the reviewer agrees to our concept of extensively describing details of our complex lidar. We hope that this encourages others to develop similar systems.*

However, the paper could enhance its impact by delving into the observational implications of these changes. Discussing how the modifications to the RMR lidar system contribute to improved observational capabilities, address specific scientific questions, or advance our understanding of atmospheric phenomena would add a valuable layer to the paper's narrative. Highlighting the potential observational impact would provide readers with a clearer sense of the practical implications and significance of the described changes.

*We thank the reviewer for this comment. We will extend the description of science questions and discuss the reasoning behind scientific as well as technical questions. The following sentence will be added in the Introduction: Most observational studies on gravity waves in the middle atmosphere below 80 km are limited to temperature variations, i.e., they describe only the potential energy of the gravity waves. Waves' kinetic energy is of at least similar significance (e.g., Geller and Gong, 2010), but remains inaccessible to most instruments. Without knowledge of the background wind, only observed, Doppler-shifted wavelengths and periods of the waves can be retrieved, but neither intrinsic periods nor vertical propagation directions can be accurately determined (Reichert et al., 2019; Strelnikova et al., 2020). Wind data from meteorological analyses introduce an unknown error into the derivation of intrinsic wave parameters, especially in the upper stratosphere and mesosphere and in highly dynamic regions like the winter polar vortex.*

*In line 393 we will add the following discussion: Even mean wind speeds above 50 km altitude in ECMWF data often deviate from the observed wind speed by 50% or more, having implications, e.g., for correct estimates of wave filtering processes. [...] Overall, this demonstrates the need for wind observations in the mesosphere for the understanding of middle atmosphere dynamics.*

*And behind line 401: This event is not at all captured in the ECMWF-IFS wind and temperature data for our site. We expect better agreement between ECMWF output and observations in summer, when the variability in the middle atmosphere is much*

*smaller. Nevertheless, this typical winter example demonstrates the need for local measurements of winds and temperatures for understanding of the dynamics in the stratosphere and mesosphere. Of course, a single site for wind/temperature observations is not sufficient for the understanding of global dynamics. We hope to foster the installation of more middle-atmosphere wind lidars through this documentation.*

The Summary will be added by this sentence (before the last sentence): *Geophysical studies of upward and downward propagating gravity waves and their intrinsic properties are currently ongoing for selected summer and winter events.*

In Section 2.4 we will explain our choice of the MQTT protocol: *Compared to that, a simple TCPIP socket connection is less versatile and always requires distinct relations between the sender and receiver of the information, which reduces the flexibility during development and operation.* The state machine concept will be justified by: *State machines are one possible concept to formalize the different tasks of the lidar measurement. They allow, again, a modular concept and retain a high degree of flexibility.*

I'd also like to suggest a correction: On line 35, it would be beneficial to check "between 30 and 80 km" and remove "(?)".

The question mark is replaced by the reference "Rüfenacht et al., 2012".

Additionally, in Figure 10, it would be helpful if the lower limits of the y-axis were provided below 45.

We are not sure about this comment. The data is shown above 35 km, which is about the lower limit of the data because of the bi-static setup described in Sec. 2.2. For lower data we would need to apply an overlap correction, which is planned for future versions of the data reduction.

With these considerations and the suggested correction, I recommend accepting the paper. Overall, it presents a valuable resource for those interested in advanced lidar systems."